# Gender Differences Associated with the Prognostic Value of BPIFB4 in COVID-19 Patients: A Single-Center Preliminary Study

**DOI:** 10.3390/jpm12071058

**Published:** 2022-06-28

**Authors:** Valentina Lopardo, Valeria Conti, Francesco Montella, Teresa Iannaccone, Roberta Maria Esposito, Carmine Sellitto, Valentina Manzo, Anna Maciag, Rosaria Ricciardi, Pasquale Pagliano, Annibale Alessandro Puca, Amelia Filippelli, Elena Ciaglia

**Affiliations:** 1Department of Medicine, Surgery and Dentistry “Scuola Medica Salernitana”, University of Salerno, Via Salvatore Allende, 84081 Baronissi Salerno, Italy; vlopardo@unisa.it (V.L.); vconti@unisa.it (V.C.); fmontella@unisa.it (F.M.); tiannaccone@unisa.it (T.I.); r.esposito86@studenti.unisa.it (R.M.E.); csellitto@unisa.it (C.S.); vmanzo@unisa.it (V.M.); ppagliano@unisa.it (P.P.); afilippelli@unisa.it (A.F.); 2Clinical Pharmacology and Pharmacogenetics Unit, University Hospital “San Giovanni di Dio e Ruggi d’Aragona”, 84131 Salerno, Italy; ginevra993@gmail.com; 3Cardiovascular Research Unit, IRCCS MultiMedica, 20138 Milan, Italy; anna.maciag@multimedica.it; 4Infectious Diseases Unit, University Hospital “San Giovanni di Dio e Ruggi d’Aragona”, 84131 Salerno, Italy

**Keywords:** COVID-19, severity, gender

## Abstract

In the ongoing global COVID-19 pandemic, male sex is a risk factor for severe disease and death, and the reasons for these clinical discrepancies are largely unknown. The aim of this work is to study the influence of sex on the course of infection and the differences in prognostic markers between genders in COVID-19 patients. Our cohort consisted of 64 adult patients (*n* = 34 men and *n* = 30 women) with PCR-proven SARS-CoV-2 infection. Further, a group of patients was characterized by a different severity degree (*n* = 8 high- and *n* = 8 low-grade individuals for both male and female patients). As expected, the serum concentrations of LDH, fibrinogen, CRP, and leucocyte count in men were significantly higher than in females. When serum concentrations of the inflammatory cytokines, including IL-6, IL-2, IP-10 and IL-4 and chemokines like MCP-1, were measured with multiplex ELISA, no significant differences between male and female patients were found. In COVID-19 patients, we recently attributed a new prognostic value to BPIFB4, a natural defensin against dysregulation of the immune responses. Here, we clarify that BPIFB4 is inversely related to the disease degree in men but not in women. Indeed, higher levels of BPIFB4 characterized low-grade male patients compared to high-grade ones. On the contrary, no significant difference was reported between *low-grade* female patients and *high-grade* ones. In conclusion, the identification of BPIFB4 as a biomarker of mild/moderate disease and its sex-specific activity would open an interesting field for research to underpin gender-related susceptibility to the disease.

## 1. Introduction 

The COVID-19 pandemic remains a global public health problem. Pre-existing comorbidities (i.e., cancer, cardiovascular and respiratory diseases, immunodeficiencies), old age, and gender have been identified as the main risk factors for the onset and progression of COVID-19 [1]. Males seem to experience more severe COVID-19, accounting for a higher proportion of hospitalizations (55%), ICU admissions (63%), and a higher case fatality rate than female patients [2,3]. By contrast, female survivors are more prone to developing long-term consequences and symptoms [4]. In this regard, sex-based differences (for the peculiar genetic, hormonal, and immune assets) in COVID-19 severity and clinical outcomes have been widely reported, but the underlying mechanisms remain poorly understood. Sex-based divergence in COVID-19 manifestations could be mainly attributable to the impact of hormonal differences on immune function. It is noteworthy that testosterone has an immunosuppressive effect, while estrogen improves immune response by influencing both innate and adaptive immune systems. For instance, estrogens mitigate pro-inflammatory cytokines released by monocytes and macrophages and positively influence T-cell development, B-cell activation, and humoral response [5,6]. Instead, testosterone reduces TNF and NO production by macrophages and decreases TLR-mediated response, lymphopoiesis, T cell differentiation and humoral response [7]. This leads the immune system to be more efficient in women rather than in men [8,9,10]. Meanwhile, immunosenescence occurring in aging contributes to enhance sexual dimorphism regarding immunity. Indeed, in adulthood, innate immune features differ from old age-related ones. For instance, NK frequency is higher in adult males, while cell phagocytic and APC activity are improved in females of the same age. Conversely, this immune asset is reversed in older males and females [10]. Instead, gender-differences concerning adaptive immune response remain almost invariable during aging. Overall, aged males show higher levels of pro-inflammatory genes (e.g., IL-6), less immune-inflammatory suppressors (e.g., IL-10), and an increased rate of decline in the numbers of B- and T-cells with respect to women [11,12]. As a result, not only females in general, but also aged females exhibit better immune machinery. Furthermore, several immunity-related genes in the X chromosome introduce another explanation to the different sex-related progression and outcomes in COVID-19. Instead, ACE2, the main virus entry receptor, is a X chromosome-encoded gene that is downregulated by estrogens that also changes according to age and sex steroids levels [13,14]. Other X-linked genes implied in viral response (i.e., those for FOXP3, XIST, TLR7/8) might guarantee major protection for females against COVID-19 [15].

Lactate dehydrogenase (LDH), high-sensitivity C-reactive protein (CRP), fibrinogen, leucocyte count, and pro-inflammatory cytokines levels are ordinary useful laboratory indicators for disease progression and severity [16]. Both age and sex physiologically affect these parameters. LDH serum levels are higher in males than females, also according to aging, while fibrinogen and CRP serum levels are lower in males and increase with age [17,18]. Moreover, leucocytes decrease with old age for men, while the opposite happens for women [19]. As adjunct value, we previously reported the role of the bactericidal/permeability-increasing fold-containing family-B-member-4 (BPIFB4) protein in mitigating the cytotoxic and pro-inflammatory effects of SARS-CoV-2, founding new prognostic significance for BPIFB4 in terms of severity for patients with COVID-19 [20]. These results were in line with our previous observations about the protective role of BPIFB4 against cardiovascular disease effects and its involvement in balancing the immune and inflammatory response. BPIFB4 is particularly expressed in olfactory epithelium, respiratory secretions and upper airways, hematopoietic and mononuclear cells holding antimicrobial and immunomodulatory properties [21,22,23]. Circulating levels of BPIFB4 are favorably associated with a higher frequency of non-classical patrolling monocytes [24], which are endowed with reparative and proangiogenic effects both in peripheral tissues and in brain vasculature. BPIFB4 has been proven to be indispensable in skewing macrophage response toward a pro-resolving M2-state (both in microglia and atherosclerotic plaque [25,26], two other key events in the immunosurveillance, tissue healing, the removal of debris, and in maintaining proper inflammatory balance. More recently the transfer of the longevity-associated variant of BPIFB4 gene, which determines its rapid enrichment both in blood and target tissues, has been described to rejuvenate the immune system and vasculature. From a functional point of view, the reduction of senescence-associated inflammation (SASP) ensured sustained NAD+ levels in the plasma of treated mice by preventing the NADase CD38 increase in F4/80+ tissue-resident macrophages [27]. This is an interesting action as the level of NAD+, a well-known nucleotide regulating cellular homeostasis that is lost as we age, contributes to metabolic dysfunction and a decline in overall fitness [28]. In this context, BPIFB4 is also able to preserve cellular homeostasis and restore proteostasis, both of which are lost with aging [23,26].

Exceptionally high circulating levels of BPIFB4 have been found in long-living individuals, the successful aging model by definition, demonstrating BPIFB4 could be involved in a heightened response against age-related and cardiovascular diseases. In this scenario, the aim of our study was to more deeply investigate the protective role of higher levels of BPIFB4 among COVID-19 patients and determine if sex may affect its prognostic value. As discussed, hereafter, BPIFB4 plasma levels can predict susceptibility to disease in men compared to women. 

## 2. Methods

### 2.1. Patients’ Recruitment

A cohort of 64 adult patients with PCR-proven SARS-CoV-2 infection was recruited to perform the study: *n* = 34 men (median age 66, range 32–90), and *n* = 30 women (median age 65, range 20–95). Peripheral blood samples were collected from each patient within 7 days from admission to the Infectious Diseases Unit of “Giovanni di Dio e Ruggi d’Aragona” University Hospital of Salerno. Samples were taken at the baseline before any kind of intervention. Concerning the occurrence of previous disease, hypertension and diabetes were among the main pathologies encountered in both group of individuals (men and women), confirming that they are two of the leading risk factors for COVID-19 complications.

As previously described, clinical laboratory analyses testing at hospital admission included: complete blood count (leucocytes, lymphocytes, platelets), mean corpuscular volume, hematocrit, hemoglobin, erythrocyte sedimentation rate, LDH, serum ferritin, D-dimer, CRP, and fibrinogen [20]. For cytokine detection and BPIFB4 level dosage, not all plasma samples were analyzed owing to hemolyzed or not adequate blood sample. Where feasible, patients were stratified in two groups according to needing oxygen or intensive care unit (ICU) admission. For each group (male and female), patients with an oxygen saturation between 90% and 94% that did not need ICU admission were considered as having mild to moderate COVID-19 (low-grade), while patients with an oxygen saturation below 90% at admission or during the hospital stay that required either noninvasive or mechanical ventilation or those that were in need of admission to the ICU were considered as having severe COVID-19 (high-grade). All participants signed an informed consent for the management of personal anamnestic data and blood samples. The study was approved by the IRCCS MultiMedica and by the internal Ethics Committee of “Giovanni di Dio e Ruggi d’Aragona” University Hospital of Salerno and was conducted in accordance with the ethical principles of the Declaration of Helsinki.

### 2.2. Cytokines Detection

IL-2, IL-4, IL-6, MCP-1, and IP-10 plasma levels were determined using a bead-based multiplex ELISA (LEGENDplexTM, Biolegend, San Diego, CA, USA). Plasma was incubated for 2 h with the beads and for 1 h with the detection antibodies, followed by 30 min incubation with SA-PE. After washing, the beads were resuspended in washing buffer and acquired using a FACS VERSE flow cytometer (BD Biosciences). The data were analyzed with LEGENDplex Data Analysis Software.

### 2.3. Enzyme-Linked Immunosorbent Assay (ELISA)

BPIFB4 plasma levels were determined using a human long palate, lung, and nasal epithelium carcinoma-associated protein 4 (C20orf186) ELISA kit (Cusabio CSBYP003694HU) while following the manufacturer’s instructions. Briefly, the plasma was incubated for 2 h at 37 °C in the coated assay microplate. After removing any unbound substances, a biotin-conjugated antibody specific for C20orf186 was added to the wells and incubated for 1 h at 37 °C. After washing, avidin-conjugated horseradish peroxidase (HRP) was added to the wells and incubated for 1 h at 37 °C. Following a wash, the substrate solution was added and consequent color development was stopped. The optical density was measured at 450 nm.

### 2.4. Statistical Analysis

In all experiments shown, statistical analysis was performed using the GraphPad prism 6.0 software package for Windows (GraphPad software). For each type of assay or phenotypic analysis, the data obtained from multiple experiments were calculated as mean ± SD and analyzed for statistical significance using appropriate tests. In the analysis of variance (ANOVA) for multiple comparisons, *p*-values < 0.05 were considered significant; * *p* < 0.05, ** *p* < 0.01, and *** *p* < 0.001. 

## 3. Results

### 3.1. Evaluation of Laboratory Parameter Profiles for Male and Female COVID-19 Patients

Increased serum levels of LDH, fibrinogen and CRP are routinely detected in COVID-19 patients [29]. Accordingly, in our cohort of patients, we previously observed higher significant levels in LDH and CRP and reduced platelet counts in severe patients (high-grade) with respect to non-severe ones (low-grade) [20]. Herein, in our cohort of *n* = 34 male (median age 66, range 32–90) and *n* = 30 female (median age 65, *range 20–95*) PCR-proven SARS-CoV-2 infected patients, the LDH, fibrinogen and CRP values were significantly higher in SARS-CoV-2 male patients as compared with the female ones (Figure 1A–C), suggesting a stronger inflammatory burst in the male group. Moreover, even if both the male and female groups showed white blood cells count in normal range (4.5–11 × 10^3^/μL), an increased leucocyte count was found for male patients with respect to the female ones (Figure 1D), reflecting different sex-related immunological impairment during SARS-CoV-2 infection [30]. These data define different profiles for the two patient groups, pointing to peripheral hematological changes in COVID-19 that are gender-related. 

### 3.2. Detection and Quantification of Cytokines in the Plasma of Male and Female SARS-CoV-2 Patients

Unbalanced pro-inflammatory cytokine secretion is associated with COVID-19 severity and mortality [31]. To assess whether a hyperinflammatory state could show different results in male and female patients, we performed a multiplex bead-based immunoassay on plasma from *high-grade* and *low-grade* patients (for both female and male). 

Overall, cytokine levels showed no significant differences among the two groups (Figure 2A–E); however, *high-grade* male patients showed higher plasma levels for pro-inflammatory IL-6, IL-2, IP-10, and MCP-1 with respect to *low-grade* male patients, as well as when compared to both *low-grade* and *high-grade* female ones. IL-4 levels were decreased in severe male patients with respect to the *low-grade* group. By contrast, the cytokine profiles of female patients were characterized by lower levels of IL-6, IL-2 and IP-10 coupled with increased levels of IL-4 in the *high-grade* group. Despite the lack of a statistical significance that probably could be achieved by expanding the recruited patient cohort, these data might suggest a gender-related inflammatory response where women might have a favorable immune profile to finely counterbalance the excessive pro-inflammatory cytokine secretion due to SARS-CoV-2 infection.

### 3.3. Differential Analysis of BPIFB4 Blood Levels in COVID-19 Patients

We have recently described that the bactericidal/permeability-increasing fold-containing family B member 4 (BPIFB4), discovered as both a longevity-associated and a host defense protein with a proved immunomodulatory activity [24,32,33], could represent a novel prognostic determinant in COVID-19. Indeed, we found lower levels of BPIFB4 in plasma from SARS-CoV-2 positive patients when compared to SARS-CoV-2 negative ones. Among SARS-CoV-2 patients, we also found more decreased BPIFB4 plasma levels in severe COVID-19 subjects with respect to *low-grade* patients, asserting that BPIFB4 levels are inversely correlated with disease severity [20]. In our cohort we observed no statistically significant differences between circulating BPIFB4 in men compared to female ones (Figure 3A). Of note, the lack of sex-based differences in BPIFB4 plasma has also been documented in control adults (median age 65, *range 26–80*), and in long-living individuals (LLI, i.e., >95 years) (Appendix A), demonstrating no sex effects on BPIFB4 plasma levels in both physiological and pathological settings. In contrast, in line with our previous results, BPIFB4 levels were inversely related to the severity of COVID-19 in the male group but not in the female group. Indeed, *high-grade* male patients showed lower levels of BPIFB4 compared to *low-grade* ones (mean 48.2 pg/mL vs. 103.27 pg/mL, *p* = 0.0503), while no differences were found between *high-grade* and *low-grade* female patients (mean 58.41 pg/mL vs. 74.1 pg/mL) (Figure 3B). No significant correlations were found between BPIFB4 and other COVID-19 inflammatory and prognostic markers (CRP, D-dimer, ferritin, etc., *data not shown*). Considering all of these findings, BPIFB4 plasma levels might represent a valuable parameter that is able to predict COVID-19 severity in a sex-related manner. 

## 4. Discussion and Conclusions

The COVID-19 pandemic has confirmed the well-established gender gap in longevity [34,35]. The virus has a negative impact on the elderly male population and sex-related alterations of the immune system in aging could be one of the critical factors affecting infection rates and disease mortality [36]. We have recently described that BPIFB4, a secreted protein belonging to innate immunity, is enriched in the CD34+ cells of LLIs [37], in the plasma and PBMCs of LLIs [24,38], and in the serum of healthy LLIs when compared to non-healthy (frail) LLIs [39] and its levels ensure a proper tuned immune response, including in COVID-19 infection [20]. Herein, the protective role of higher levels of BPIFB4 in men compared to women could help to clarify the sex effects on susceptibility to the disease. Concerning the putative mechanism, it is noteworthy that the BPIFB4 gene can exhibit different polymorphisms [37], thereby regulating its abundance and association with respect to longevity [38]. In this context, it is not surprising to detect gender differences in genetic factors, such as for foxo3A polymorphism correlation with longevity as found in the Southern Italian Centenarian Study (SICS), showing a protective effect only in males [40]. We think that males that are frailer and are exposed to more diseases have shorter life spans when compared to females, and thus are the best target for protective genetic modifiers like BPIFB4. We have already reported that higher BPIFB4 levels play a protective role in many disease conditions which are particularly associated with frailty in old males, such as diabetes [41], atherosclerosis [26], endothelial dysfunction and heart failure [submitted], in which gene therapy approaches leading to an improvement of its function may represent a potential therapeutic pathway. In addition, a variant of BPIFB4 gene that has been found to be associated with higher protein levels can selectively counteract some features of immunesenescence in vitro and in vivo [27]. Immunesenescence, the phenotypic and functional impairment of the immune responses occurring as we age, is a predisposing risk factor for the development of COVID-19 [42]. As immunesenescence is more profound in males when compared to females, the senotherapeutic effects of BPIFB4 would occur preferentially in individuals with a high senescent cell burden (e.g., males) and this would result in increased disease severity in a gender-related manner.

Overall, the specific prognostic value of BPIFB4 on COVID-19 severity for males but not females may not only corroborate the well-established sex-related effects of immune mediators, but may also be considered as an intrinsic evolutionary event aimed to counteract the male disadvantage for life expectancy.

## Figures and Tables

**Figure 1 jpm-12-01058-f001:**
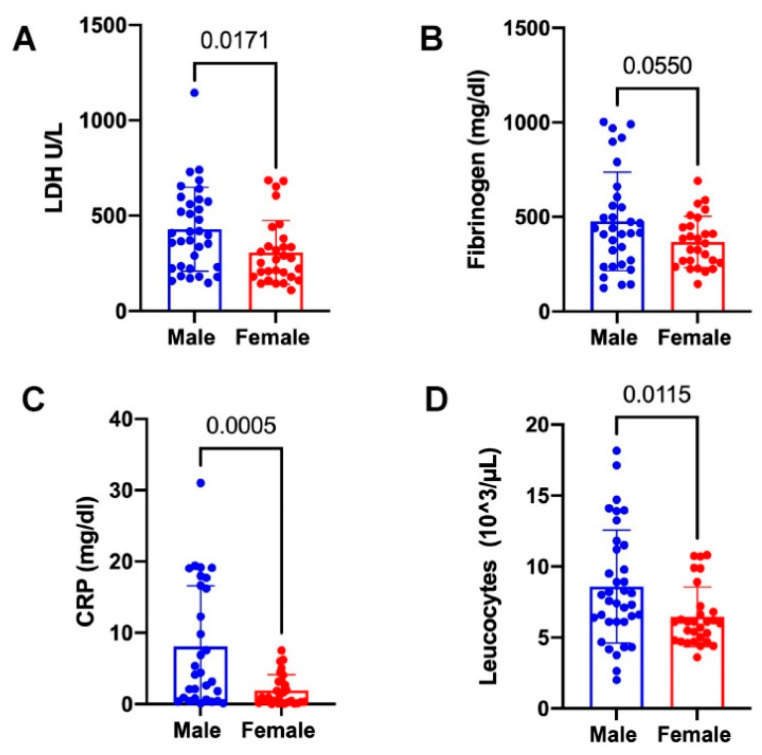
Selected laboratory parameter profiles for male and female COVID-19 patients. Significant differences in the laboratory levels of (**A**) LDH, (**B**) fibrinogen, (**C**) CRP, and (**D**) leucocyte counts in *n* = 34 males, with respect to *n* = 30 female COVID-19 patients. Statistical evaluation was carried out via one-way ANOVA. Results are expressed as the mean ± SD. The individual *p*-values are shown.

**Figure 2 jpm-12-01058-f002:**
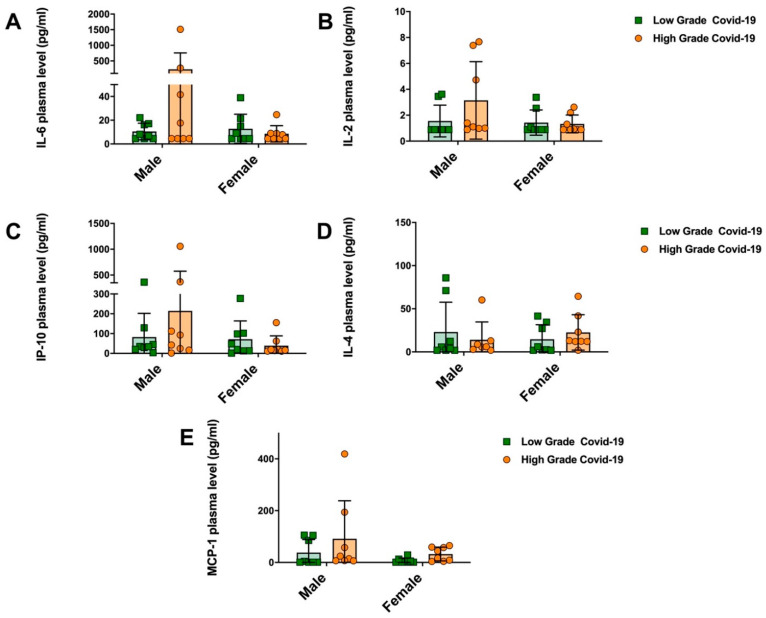
Cytokine plasma levels in male and female SARS-CoV-2 patients. Plasma from *n* = 16 male and *n* = 16 female SARS-CoV-2 patients were collected and (**A**) IL-6, (**B**) IL-2, (**C**) IP-10, (**D**) IL-4 and (**E**) MCP-1 cytokine levels were determined using a bead-based multiplex ELISA. Each group was stratified into *n* = 8 low grade and *n* = 8 high grade COVID-19 patients. Results are expressed as mean ± SD. No significant *p*-values were found.

**Figure 3 jpm-12-01058-f003:**
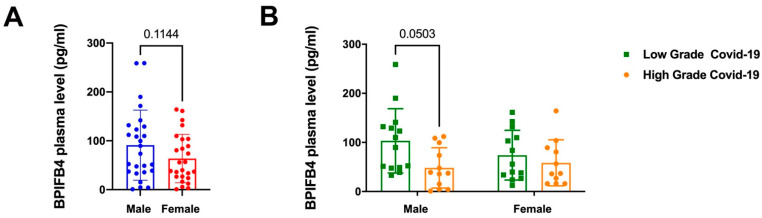
Circulating levels of the BPIFB4 protein from male and female SARS-CoV-2 positive subjects. (**A**) ELISA quantification of BPIFB4 levels in plasma from *n* = 26 male and *n* = 26 female COVID-19 patients expressed as mean ± SD. Statistical evaluation was carried out with one-way ANOVA. The individual *p*-values are shown. (**B**) Data obtained from ELISA quantification of BPIFB4 levels from both male and female SARS-CoV-2 positive group were analyzed by layering in *n* = 14 low grade and *n* = 12 high grade patients.

## Data Availability

The authors confirm that the data supporting the findings of this study are available within the article and its supplementary materials. Raw data are available from the corresponding authors, [EC], upon reasonable request.

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
