# Peer review of "Gender Differences Associated with the Prognostic Value of BPIFB4 in COVID-19 Patients: A Single-Center Preliminary Study"

_jpm, 2022, doi:10.3390/jpm12071058_

Round 1
Reviewer 1 Report
see attached

Author Response
The manuscript titled Gender Differences associated with the prognostic value of BPIFB4 in COVID-19 patients” by Lopardo et. al is a descriptive study, aimed at characterization of prognostic markers for COVID-19 in the serum. The paper identifies the factor BPIFB4 to differentially expressed in high grade versus low grade COVID-19 male patients.
The manuscript is suitable for publication. I would recommend the following minor corrections:
We thank the reviewer for his/her positive remarks and we are pleased to know that he/she considered our “manuscript suitable for publication”. We also thank the reviewer for his/her meaningful suggestions on how to improve our scientific report.
- The 4th paragraph in the abstract is not very clear and should be re-written, especially the part starting with “while no potential..”
Sure, as required by the reviewer, we’ve better detailed the sentence.
- There is an extra space in page 4 last line
Ok
- The last paragraph of the introduction should state the aim of the study and the main finding
Sure, done.
- Page 11, the sentence starting with “Even if…”, please rephrase in a more scientific way
Sure, done.
- The paper is looking into a single parameter analysis at a time, please discuss the potential of combined, several parameters for statistically significant differentiation and correlation with disease state
We thank the reviewer for his/her request of explanation
On the basis of laboratory test, no significant correlations were found between BPIFB4 and other COVID-19 inflammatory and prognostic markers (CRP, D-dimer, ferritin, etc.) as already reported elsewhere (Ciaglia E. J Gerontol A Biol Sci Med Sci. 2021 Sep 13;76(10):1775-1783. doi:10.1093/gerona/glab208.) This finding if one hand suggests that other parameters need to be co-investigated, on the other hand corroborates the prognostic relevance of BPIFB4 levels as biomarker in COVID-19. We’ve better discussed this point at the end of results section
Reviewer 2 Report
In this manuscript, authors analyzed the prognostic significance of BPIFB4 in gender-dependent differences in the course of Covid 19 infections in a small cohort of patients. It is interesting to study the novel indication, however the cohort is very small for a meaningful outcome. Below are concerns that would improve the manuscript.
n Authors should study if BPIFB4 levels would indicate the terminal outcome in these patients (survival and death)
n The mechanism of BPIFB4 is not very clear from the discussion. The authors can provide some more insights on how the BPIFB4 can alter the immune response in males but not in females.
n Abstract --- “While no potential sex-related modification of total BPIFB4 plasma level was detected, in men, but not in women, BPIFB4 was inversely related to the disease degree; indeed, higher levels of BPIFB4 characterized low-grade male patients compared to high-grade ones.” – This sentence is too long to difficult to understand. Please revise it to simplify it.
n Page 4 (Introduction) – “LDH serum levels are higher in male than female also according to aging, while fibrinogen and PCR serum levels are lower in male and increases with age..” – Here PCR is misspelled. It should be CRP.
n Supplementary figure 1 – Please include figure legend and also include the total N numbers for each group. It is unclear from the figure.
Author Response
Reviewer: 2
Comments to the Authors
In this manuscript, authors analyzed the prognostic significance of BPIFB4 in gender-dependent differences in the course of Covid 19 infections in a small cohort of patients. It is interesting to study the novel indication, however the cohort is very small for a meaningful outcome. Below are concerns that would improve the manuscript.
We thank the reviewer for his/her positive remarks and we are pleased to know that he/she considered our data “interesting to study as novel indication”. We also thank the reviewer for his/her meaningful suggestions on how to improve our scientific report.
-Authors should study if BPIFB4 levels would indicate the terminal outcome in these patients (survival and death)
We thank the reviewer for his/her request of explanation. Unfortunately only data from the first sampling at admission were used for our purpose. Indeed, the blood samples were taken at the baseline before any kind of intervention and follow-up data were not available.
As detailed in M&M section, we had only clinical laboratory analyses testing at hospital admission including complete blood count (leucocytes, lymphocytes, platelets), mean corpuscular volume, hematocrit, hemoglobin, erythrocyte. Based on these parameters we were uniquely able to distinguish severe patients (characterized by higher significant levels in lactate dehydrogenase (LDH) and C-reactive protein (CRP) and reduced platelets count)from the non-severe ones.
-The mechanism of BPIFB4 is not very clear from the discussion. The authors can provide some more insights on how the BPIFB4 can alter the immune response in males but not in females.
This is an interesting question. We thank the reviewer for this. As already discussed, we think that's, as other genetics modifiers, the BPIFB4 protective levels in male but not in female can represent an evolutionary event aimed to counteract the male disadvantage in life expectancy.
In addition, a variant of BPIFB4 gene, found associated with higher protein level, selectively counteracted some features of immune senescence in vitro and in vivo.,
Immunesenescence, the phenotypic and functional impairment of the immune responses occurring as we age, is a predisposing risk factor for the development of COVID-19. As immunesenescence is more profound in males when compared to females, the senotherapeutic effects of BPIFB4 would occur preferentially in individuals with a high senescent cell burden (males) and result beneficial to mark disease severity. Of course, the new point of discussion was inserted in the revised version of the manuscript.
n Abstract --- “While no potential sex-related modification of total BPIFB4 plasma level was detected, in men, but not in women, BPIFB4 was inversely related to the disease degree; indeed, higher levels of BPIFB4 characterized low-grade male patients compared to high-grade ones.” – This sentence is too long to difficult to understand. Please revise it to simplify it.
Sure, as required by the reviewer, we’ve better detailed the sentence.
n Page 4 (Introduction) – “LDH serum levels are higher in male than female also according to aging, while fibrinogen and PCR serum levels are lower in male and increases with age..” – Here PCR is misspelled. It should be CRP.
OK, done
n Supplementary figure 1 – Please include figure legend and also include the total N numbers for each group. It is unclear from the figure.
Sure, the reviewer is right. A complete figure legend for Supplem. Fig.1 has been included in the revised text.